# The Phosphoproteome of the Rd1 Mouse Retina, a Model of Inherited Photoreceptor Degeneration, Changes after Protein Kinase G Inhibition

**DOI:** 10.3390/ijms24129836

**Published:** 2023-06-07

**Authors:** Jiaming Zhou, Charlotte Welinder, Per Ekström

**Affiliations:** 1Ophthalmology, Department of Clinical Sciences Lund, Faculty of Medicine, Lund University, 221 00 Lund, Sweden; jiaming.zhou@med.lu.se; 2Mass Spectrometry, Department of Clinical Sciences Lund, Faculty of Medicine, Lund University, 221 00 Lund, Sweden; charlotte.welinder@med.lu.se

**Keywords:** retinal degeneration, phosphoproteome, mass spectrometry, organotypic retinal explant culture

## Abstract

Retinitis pigmentosa (RP) is a frequent cause of blindness among the working population in industrial countries due to the inheritable death of photoreceptors. Though gene therapy was recently approved for mutations in the RPE65 gene, there is in general no effective treatment presently. Previously, abnormally high levels of cGMP and overactivation of its dependent protein kinase (PKG) have been suggested as causative for the fatal effects on photoreceptors, making it meaningful to explore the cGMP-PKG downstream signaling for more pathological insights and novel therapeutic target development purposes. Here, we manipulated the cGMP-PKG system in degenerating retinas from the *rd1* mouse model pharmacologically via adding a PKG inhibitory cGMP-analogue to organotypic retinal explant cultures. A combination of phosphorylated peptide enrichment and mass spectrometry was then applied to study the cGMP-PKG-dependent phosphoproteome. We identified a host of novel potential cGMP-PKG downstream substrates and related kinases using this approach and selected the *RAF1* protein, which may act as both a substrate and a kinase, for further validation. This showed that the *RAS/RAF1/MAPK/ERK* pathway may be involved in retinal degeneration in a yet unclarified mechanism, thus deserving further investigation in the future.

## 1. Introduction

The hereditary disease Retinitis pigmentosa (RP) is a frequent cause of blindness among the working population in industrial countries [1]. There is no effective treatment available at present [2], with the notable exception of the approved gene therapy for mutations in the RPE65 gene [3]. RP results in vision loss via fatalities of the light-sensing photoreceptors, which are categorized as rods and cones. The degeneration pattern starts with the death of rods that cause night blindness, which is followed by the degeneration of cones [4] and hence a loss of general vision. The *rd1* mouse model, with the gene mutation of the beta subunit of the enzyme phosphodiesterase-6, is characterized by an abnormally high level of photoreceptor cyclic guanosine monophosphate (cGMP) and a rapid loss of these cells [5]. The degeneration of photoreceptors in this phenotype has been demonstrated with an increasing decline in outer retinal thickness and overall amplitude reduction in the waveform via spectral domain optical coherence tomography [6] and electrophysiological changes [7], respectively. Although the details on how the high level of cGMP affects the photoreceptors are still not known, previous studies have suggested that at least part of the death-promoting impact on photoreceptors could be exerted via increased activity of cGMP-dependent protein kinase (PKG) probably causing over-phosphorylation within photoreceptors [8,9]. It would thus be most helpful if this could be studied more closely.

There are two forms of PKG, *PKG1*, and *PKG2*, with *PKG1* having two isoforms, PKG1alpha and PKG1beta [10]. cGMP-PKG functions by phosphorylation of certain serine and threonine residues in a number of biological targets [10]. The importance of this so-called cGMP-PKG system has been recognized for its essential role in gene regulation in diverse tissues [11,12] and also for its involvement in the cell death mechanism [13]. Despite the deepening insights regarding this system, we are, as indicated above, still confronted with one unsolved problem, namely, what are the actual details of the downstream signaling of the cGMP-PKG system that could affect the RP progression?

Certain analogs of cGMP, a category of chemically modified versions of the native cGMP [14,15,16], exert considerable inhibition or activation of PKG isoforms. This has been taken advantage of when testing potential treatments, where the systemic administration of a selected liposome-formulated cGMP analog with PKG inhibitory properties was shown to protect RP retinas from retinal degeneration and to preserve the photoreceptor function [9]. To reveal the downstream signaling practically, one may thus manipulate the retinal cGMP-PKG system pharmacologically via the addition of relevant cGMP-analogues during organotypic retinal explant culturing and, subsequently, explore the cGMP-PKG downstream pathways with a phosphoproteomic analysis, in analogy with how the effects of PKG manipulation on the retinal transcriptome has been studied [17]. Together this should be beneficial for studying the cGMP-PKG-dependent targets.

In this study, we have therefore used a PKG inhibitor on *rd1* retinal explants, and via a phosphoproteomic analysis, which combined phosphopeptide enrichment with mass spectrometry (MS) [18], compared the resulting phosphopeptide pattern with that of the untreated counterparts. The selected lists of altered phosphorylated sites after PKG inhibition were then further analyzed via appropriate bioinformatics methods to reveal the downstream pathways that were potentially connected to RP advancement. To study the possible effectors of the cGMP-PKG system in RP, we subsequently picked up several phosphorylated proteins for further validation of phosphorylation. Our results show a number of phosphorylated sites that are potentially controlled by cGMP-PKG in the *rd1* retina and, as such, provide more insights into the retinal degeneration mechanisms in RP.

## 2. Results

### 2.1. Quantitative Phosphoproteomics of Retinal Explants with PKG Inhibition

In total, 17,915 phosphorylated sites and 3405 retinal proteins were identified (Figure 1A); among these, 10,352 phosphosites from 2214 proteins were selected for further analysis, according to the defined criteria (detailed in Section 4). Comparison via Student’s *t*-test showed that 299 phosphosites of 108 proteins had decreased phosphorylation, while 184 phosphosites of 83 proteins had increased phosphorylation in retinas with PKG inhibition for 2 h compared to their untreated counterparts (Figure 1B, Table 1; see Appendix A for a full list, with phosphorylation sites, fold changes, etc.). For the longer treatment, 4 days, 98 phosphosites of 46 proteins were observed with decreased phosphorylation, while 265 phosphorylation sites of 130 proteins with increased phosphorylation. (Figure 1C, Table 2, and Appendix A for full list).

There were five overlapping proteins with decreased phosphorylation after both 2 h (short term) or 4 days (long term) PKG inhibition. (Figure 1D and Table 1; overlapping proteins highlighted in yellow in Appendix A). For proteins with increased phosphorylation, there were 16 overlapping proteins after short and long term PKG inhibition (Figure 1E and Table 2; overlapping proteins highlighted in yellow in Appendix A). When we focused on individual phosphorylation sites in the peptides of the overlapping proteins, there were two proteins with decreased phosphorylation that were represented by the very same, specific phosphorylation sites at both time points, and three proteins from those with increased phosphorylations (Table 1 and Table 2; plus labeled with green in Appendix A).

### 2.2. Potential cGMP-PKG Dependent Biological Pathways

A protein annotation in the Kyoto Encyclopedia of Genes and Genomes (KEGG), with the targets mentioned above, was performed to address the biological pathways in which the cGMP-PKG-dependent substrates were involved. Proteins with decreased phosphorylated sites after PKG inhibition were annotated in RNA transport, Spliceosome, and Rap1 signaling pathway (Figure 2A). A part of these down-regulated sites after PKG inhibition, such as the sites in protein kinase cGMP-dependent 1 (*PKG1*), general transcription factor IIi (*GTF2I*), and protein phosphatase 1 catalytic subunit alpha (*PPP1CA*), were involved in the cGMP-PKG pathway. This supported that PKG signaling in explants was appropriately inhibited during culturing. By contrast, the proteins of the increased phosphorylation were mostly related to pathways connected with proteoglycans, ErbB signaling, and tight junctions (Figure 2B). In this manner, we identified potential signal pathways that may function under cGMP-PKG regulation in photoreceptors undergoing degeneration.

### 2.3. Upstream Kinase Analysis

To predict the upstream kinases for the altered phosphorylations identified above, the regulated phosphoproteins were analyzed via annotation to the online tool Phosphomatics V2 (https://phosphomatics.com/). The outcomes are given as kinome trees (Figure 3A,B) and as lists (Appendix A). For the 2 h treatment, the algorithms suggested a total of 66 potential upstream kinases, 32 with decreased activity and 34 with increased activity (Figure 3A, Appendix A). Among these, 7 kinases in the category of proposed decreased activity, and 11 of the increased ones phosphorylate their substrates in a site-specific manner (Appendix A).

For the 4 days treatment, the number of suggested upstream kinases reached 68, of which 8 had decreased activity and 60 had increased activity (Figure 3B, Appendix A). In this case, the site-specific ones were represented by three and five kinases of those with proposed decreased and increased activity, respectively (Appendix A). The kinases that had a consistent reaction to the treatment between the two time points were summarized (Table 3). Two kinases gave indications of having reduced activity after both time points, namely cyclin-dependent kinase 1 (*CDK1*) and cAMP-dependent protein kinase catalytic subunit alpha (*PRKACA*), whereas 10 kinases came up as having increased activity at both time points. Of the latter, four kinases were related to the mitogen-activated protein kinase (*MAPK*) family (Table 3). There were also some kinases that displayed opposite reactions to the treatment after 2 h and 4 days (Table 3), such as *CDK2*, *CDK7*, *MAPK3*, and more. These are not considered further in this report. To sum up, according to the upstream kinase analysis with the selective differentiated phosphoproteins mentioned above, a series of kinases regulated by cGMP-PKG were revealed to potentially function during retinal degeneration.

### 2.4. Selection of RAF-1 for Further Validation

To gain more insights in the cGMP-PKG network, one target was selected from the upstream kinase analysis for further validation. Since the abnormally high level of cGMP in the photoreceptors is regarded as a disease driver of retinal degeneration [19], we assumed that photoreceptor-specific proteins have an essential role in retinal dysfunction. For this purpose, the RAF proto-oncogene serine/threonine–protein kinase (*RAF1*) was selected for additional analysis. As mentioned above, several of the upstream kinases with increased activities belonged to the *MAPK* family. The rationale for selecting RAF is thus that *MAPK1* is expressed in photoreceptors and is linked to photoreceptor death, at any rate in Drosophila [20], and also that RAS and its downstream pathway component *RAF1* are required to satisfy photoreceptor differentiation [21]. In addition, *RAF1* was proposed as having higher kinase activity after 2 h of PKG inhibition (Appendix A). Therefore, as part of the well-established *Ras/Raf-1/MAPK1* signaling, the role of *RAF1* during retinal degeneration and its connection with the cGMP-PKG system was of interest to analyze further. Other kinases identified in this study appear to be good subjects for deeper investigation in the future.

To see whether *RAF1* is specifically expressed within photoreceptors, retinas from either *rd1* or wt animals were immunostained for *RAF1*. In wt, *RAF1* was expressed predominantly in the photoreceptor segments. This was also true for *rd1*, which in addition, displayed elevated *RAF1* staining in many parts of the *rd1* outer nuclear layer (ONL), where the photoreceptor cell bodies and processes reside (Figure 4A–C). This supports that *RAF1* can be seen as a relatively photoreceptor-specific kinase and that it could be involved in photoreceptor degeneration.

Next, the potential link between *RAF1* expression and the cGMP-PKG system was investigated at the histological level. This was achieved via pharmacological PKG inhibition in retinal explants during culturing (the same culturing paradigms as above), followed by an analysis of possible *RAF1* alterations in photoreceptors after treatment. Figure 5 shows that the *RAF1* expression was higher in the ONL after PKG inhibition, regardless of the different lengths of treatment.

To see if the change in *RAF1* expression was in any way also reflected in the phosphorylation state of *RAF1*, the PLA technique for the combination of *RAF1* protein and phosphorylated serine [22] to study the *RAF1* phosphorylation profile was applied. The PLA uses antibodies against two different antigens, as well as certain secondary antibodies containing complimentary “primers” of a reaction product. If the two antigens are in close proximity (<40 nm), the method then allows a reaction product to be formed. Thus, if the RAF antibody is in close contact with the phosphoserine antibody, it is likely that both antibodies sit on the same protein, namely *RAF1*, and thus identify phosphorylated *RAF1*. We focused on the *RAF1* phosphorylation in ONL since *RAF1* was higher expressed in the *rd1* ONL (Figure 4) and found an increased level of PLA signal within the ONL of *rd1* (Figure 6). This indicated a greater extent of *RAF1* phosphorylation in retinas undergoing degeneration.

The next question was if the *RAF1* phosphorylation could be affected by the manipulation of the cGMP-PKG system during culturing. We observed that *RAF1* phosphorylation was higher in the ONL of retinal explants being treated with PKG inhibitor for 2 h (Figure 7A,B,E), while no difference to the untreated counterpart was noticed after 4 days of treatment (Figure 7C–E). This was consistent with our discovery in the upstream kinase analysis that a higher *RAF1* activity was identified in explants with 2 h of PKG inhibition but not in the 4 days treatment (Figure 3A,B and Appendix A). The combined data thus clearly indicate the participation of *RAF1* activities in retinal degeneration during the 2 h of PKG inhibition, albeit in an unclarified mechanism. It is reasonable to deduce that different lengths of PKG inhibitory treatment affect the *RAF1* activities, such that RAF may be only transiently affected by the change in PKG activity, but that this still results in effects downstream to *RAF1* itself. Note that the overwhelming amount of the phosphorylated *RAF1* resided in the ONL and the photoreceptor segments, again suggesting a potential role of *RAF1* during retinal degeneration, though further investigation is required to specify whether *RAF1* phosphorylation promotes neurodegeneration or neuroprotection in photoreceptors.

## 3. Discussion

Despite gene therapy targeting the mutated *RPE65* gene having been developed to treat RP [3], other effective treatments are required because of the genetic heterogeneity of the disease [23]. The cGMP-PKG system [5], which is regarded as one of the RP disease drivers, is the potential molecular target for a novel therapeutic strategy development for this disease. In this context, the cGMP-PKG system is supposed to function via the phosphorylation of retinal substrates, including in photoreceptors [8,9]. It is, therefore, likely that cGMP-PKG-dependent phosphoproteomics will provide more insights into the details of retinal degeneration. In this case, MS-based phosphoproteomics techniques, with extensive application in biological research [24], would be a proper method to investigate the cGMP-PKG downstream signaling. Technically, one option is to apply immunoprecipitation for phosphoprotein enrichment prior to MS detection. However, due to the limitation of commercially available phospho-selective antibodies, this approach has not provided promising results in phosphoprotein enrichment [25] and is, therefore, unsuitable for high throughput phosphoproteomic study. Another widely applied strategy is the use of immobilized metal affinity chromatography, as in the current study. Though the non-specificity has been noticed for this method, the continuous optimization of the protocol has made it frequently applied in phosphoproteomic research [26], including here.

Our study has three clear outcomes: (1) it identified potential retinal PKG substrates that could be involved in retinal degeneration, (2) it gave information on the kinase and pathway networks in operation during the degenerative events, and (3) it showed a correlation between the cGMP-PKG system and the *RAS/RAF/MAPK/ERK* pathway in this situation.

The upstream kinase and KEGG analyses both picked up PKG or the cGMP-PKG signaling pathway as being affected by the treatment, which corroborated the usefulness of the approach. In addition, the two treatment regimes, i.e., 2 h or 4 days of PKG inhibition, gave data on proteins whose phosphorylation was decreased as well as such that displayed increased phosphorylation in either or both of these regimes.

The assumption that increased phosphorylation by PKG is part of the degeneration mechanism makes the proteins with reduced phosphorylation at both time points of PKG inhibition particularly interesting with respect to a pathologic involvement. Among the five proteins of this category, we noticed DNA Topoisomerase II Beta (*TOP2B*) and capicua transcriptional repressor (*CIC*). Interestingly our previous transcriptome study revealed the association between cGMP-PKG and these two targets; hence, there is reason to keep *TOP2B* and *CIC* as candidates for links between PKG activity and events during retinal degeneration. Proteins that instead showed persistent increased phosphorylation after PKG inhibition are, by contrast, more likely to be dependent on kinases under direct or indirect negative regulation by PKG and were represented by as many as 16 proteins. Some deserve to be mentioned here, such as ankyrin 2 (*ANK2*), with a possible role in maintaining ion balance, via anchoring of ion transporters to the photoreceptor cell membrane [27]. Proteins of the microtubule-associated protein (*MAP*) family, including *MAP1B*, *MAP2*, and *MAP4*, were also seen. *MAPs* could be critical for photoreceptors, and at least one of these may associate with retinal degeneration [28,29].

The upstream kinase analysis indicated integration of the cGMP-PKG pathway with other kinases. With the same rationale mentioned above, we noticed *CDK1* that has a potentially lower activity after 2 h and 4 days of PKG inhibition. Given that we recently demonstrated its expression in degenerating *rd1* photoreceptors, as well as that this was reduced by PKG inhibition [30], a possible interpretation would be that CDK1 expression is somehow affected by a cGMP-PKG regulated phosphorylation. By contrast, several kinases with potential higher activities after PKG inhibition were from the *MAPK* family, which appear connected with the degeneration, including in a PKG-related way.

The *RAS/RAF1/MAPK/ERK* pathway, with its diverse biological functions, is one of the most extensively studied signal transduction routes [31]. The role of *RAF1* has been studied in diabetic retinopathy [32], and here we provide insights into how *RAF1* may connect with RP, which has seldom been noticed previously. In wt retinas, the predominant expression of *RAF1* was in the photoreceptor segments, pointing to it being a photoreceptor-specific kinase. The increased expression and phosphorylation of *RAF1* in the *rd1* compared to wt ONL revealed that it may participate in the degeneration of the photoreceptors, not the least since the above suggested increased activation of several MAPKs.

However, neither its increased expression nor the increased phosphorylation specifies whether *RAF1* works for or against the degeneration. Our results of PKG inhibition in *rd1* explants provided more clues, though, since *RAF1* expression increased after both lengths of treatment and more phosphorylated *RAF1* was seen after 2 h of PKG inhibition. Given that retinal degeneration benefits from such PKG inhibition [8,9], *RAF1* may exert neuroprotective effects during photoreceptor death. Had it been the opposite, that the expression and phosphorylation state of *RAF1* drives the degeneration, both these parameters would have been expected to decrease after PKG inhibition. Yet the situation is not straightforward since phosphorylation of *RAF1* may lead to either its inhibition or activation, depending on the actual serine site [33], which we were not able to discriminate between here. While highly speculative, the suggested activation of *RAF1* in the upstream kinase analysis could mean that the degeneration initiates a protective program involving *RAF1* activity but that PKG counteracts the same by blocking the activity, probably via other kinase or even phosphatase activities.

In conclusion, our study shows an intricate pattern of changes with respect to protein phosphorylations after inhibition of PKG. It appears likely that several of these results, including those connected to the RAS/*RAF1*/MAPK/ERK pathway, relate to the yet unclarified mechanisms of inherited retinal degeneration and, as such, will aid in designing future investigations in this area.

## 4. Materials and Methods

### 4.1. Animals

The C3H/*rd1*/*rd1* (*rd1*) and control C3H wild-type (wt) were kept and bred in-house under standard white cyclic lighting, with free access to food and water [34]. The animals were used irrespective of sex since it is practically difficult to discriminate at the early age when they were sacrificed in our protocol. The day of birth of the animal was considered as postnatal 0 (P0), with the day following this considered as P1, etc.

### 4.2. Organotypic Retinal Explant Culture

Retinas from P5 mice were used to generate explants according to our standard protocol [17]. Inserts with the explants were put into six-well culture plates with 1.5 mL serum-free medium in each well. Plates were incubated at 37 °C with a 5% CO_2_ atmosphere, and the medium was replaced every two days. No additions were made to the cultures for the first 2 days. One group of the cultures, which at this point were at an age corresponding to P7, were then exposed to 50 µM Rp-8-Br-PET-cGMPS (PKG inhibitor; Biolog, Bremen, Germany, Cat. No.: P 007) for the following 4 days, with the endpoint thus equivalent to P11 (n = 4), with their corresponding controls (n = 4) receiving the same amount of distilled water. Another group (n = 4) was exposed to the Rp-8-Br-PET-cGMPS with the same concentration but only for 2 h before the end of the protocol, with controls (n = 4) similarly exposed to distilled water. All retinal explants were either collected for protein extraction, followed by phosphoproteomic analyses. In parallel, another four groups of *rd1* explants, reaching a total of 13 explants, were generated. One group of retinas (n = 4) was exposed to the 50 µM Rp-8-Br-PET-cGMPS for 4 days, while their corresponding untreated controls (n = 3) received the same amount of distilled water. The remaining two groups included one (n = 3) that was treated with 2 h of PKG inhibition with the same concentration and one with its untreated peers (n = 3). These explants were fixed, sectioned, and used for microscopy-based studies.

Animals were randomly assigned to the experimental groups, and the experimenters were aware of the conditions of the animals during retinal explant culture. No statistical method was applied to pre-determine the sample size of the experimental groups. The sample size was chosen based on the relevant literature in the field as well as on ethical considerations. The study was not pre-registered. The study was exploratory without exclusion criteria being pre-determined.

### 4.3. Sample Preparation for MS

A total of 16 retinal explants were used for MS measurement, and 4 explants were used per group. The experimenter was unaware of the explant’s group during experimentation. Each explant was homogenized separately in buffer (50 mM Tris-HCl, 50 mM NaCl, 1 mM EDTA, 5 mM NaH_2_PO_4_, 1 mM DL-Dithiothreitol (DTT)), supplemented with phosphatase inhibitors (Lot No, 33041800, Roche, Basel, Switzerland, 1 tablet per 10 mL buffer) using a homogenizer (Knotes Glass Company, Vineland, NJ, USA). The homogenate was then centrifuged at 10,000× *g* for 5 min at 4°C. The soluble fraction was collected after centrifugation, with the concentration measured by Bio-Rad Protein Reagent Assay Kit (Cat. No.: #5000113, #5000114, #5000115, Bio-Rad, Hercules, CA, USA).

For each separated sample, proteins were reduced with DTT to a final concentration of 10 mM and heated at 56 °C for 30 min, followed by alkylation with iodoacetamide for 30 min at room temperature in the dark to a final concentration of 20 mm. Subsequently, samples were precipitated with ice-cold ethanol overnight at −20 °C followed by centrifugation at 14,000× *g* for 10 min. The pellets were resuspended in 100 mM ammonium bicarbonate and sonicated for 20 cycles of 15 s on and 15 s off, using a Bioruptor (Diagenode, Denville, NJ, USA). Digestion was performed by adding trypsin (Sequencing Grade Modified Trypsin, Part No. V511A, Promega, Madison, WI, USA) in a ratio of 1:50 to the samples and incubated overnight at 37 °C. The digestion was stopped by the addition of 5 µL 10% trifluoroacetic acid (TFA).

The Pierce High-Select Fe-NTA Phosphopeptide Enrichment Kit (Cat. No.: A32992; Thermo Fischer Scientific, Waltham, MA, USA) was used to enrich phosphopeptides according to the manufacturer’s protocol. The phosphopeptides were run in a Speed Vac to dryness and resolved in 2% acetonitrile (ACN) and 0.1% TFA to a peptide concentration of 0.25 µg/µL.

### 4.4. MS Acquisition and Analysis

The peptide analyses were performed on a Q Exactive HFX mass spectrometer (Thermo Scientific) connected to an EASY-nLC 1200 ultra-high-performance liquid chromatography system (Thermo Scientific). Peptides, 1 µg, were separated on an EASY-Spray column (Thermo Scientific; ID 75 μm × 50 cm, column temperature 45 °C) operated at a constant pressure of 800 bar. A two-step gradient of buffer B (80% acetonitrile, 0.1% formic acid) in buffer A (aqueous 0.1% formic acid) was applied at a flow rate of 300 nL min^−1^. In the first step, a gradient of 10 to 30% of buffer B was run for 90 min, followed by a 30 to 45% gradient of buffer B in 20 min. One full MS scan (resolution 60,000 @ 200 *m*/*z*; mass range 350–1400 *m*/*z*) was followed by MS/MS scans (resolution 15,000 @ 200 *m*/*z*). The precursor ions were isolated with 1.3 *m*/*z* isolation width and fragmented using higher-energy collisional-induced dissociation at a normalized collision energy of 28. Charge state screening was enabled, and singly charged ions as well as precursors with a charge state above six were rejected. The dynamic exclusion window was set to 10 s. The automatic gain control was set to 3 × 10^6^ for MS and 1 × 10^5^ for MS/MS, with ion accumulation times of 45 ms and 60 ms, respectively. The intensity threshold for precursor ion selection was set to 1.7 × 10^4^.

The raw DDA data were analyzed with Proteome Discoverer™ Software (Version 2.3, Thermo Fisher Scientific). Peptides were identified using both SEQUEST HT and Mascot against the UniProtKB mouse database (UP000000589 plus isoforms). The search was performed with the following parameters applied: static modification: cysteine carbamidomethylation and dynamic modifications: N-terminal acetylation. Phosphorylation (S, T, Y, for serine, threonine, and tyrosine, respectively) was set as a variable for the phosphopeptide analysis. Precursor tolerance was set to 10 ppm and fragment tolerance to 0.02 ppm. Up to 2 missed cleavages were allowed, and Percolator was used for peptide validation at a q-value of a maximum of 0.05. Peptides with different amino acid sequences or modifications were considered unique. Extracted peptides, with a quality q-value lower than 0.05, and modification of S, T, and Y phosphorylation detected were used to identify and quantify them by label-free relative quantification. The extracted chromatographic intensities were used to compare peptide abundance across samples.

The MS results were processed via Perseus software (version 1.6.0.7) [35]. The protein intensities were log2 transformed. Selective criteria were defined as follows: among 4 replicates of 4 conditions (16 samples total), only a peptide with the missing value of less than 30% in total (less than 5 samples with missing value) would be selected, and the missing values were replaced from a normal distribution was performed through data imputation by using the following settings: width 0.3 and downshift 0. Further bioinformatics analysis of these processed data was performed via the web-based tool Phosphomatics (https://phosphomatics.com/; [36]). In Phosphomatics, two-Sample Student’s *t*-test (two-tailed) was performed to compare phosphorylated site levels between the *rd1* explants with 2 h of PKG inhibition vs. untreated control and 4 days of PKG inhibition vs. untreated controls. A *p*-value of 0.05 was defined as the cutoff. Two lists of the phosphorylated sites identified previously were used to perform a further upstream kinase analysis in Phosphomatics. The biological pathways that might be affected between *rd1* untreated and *rd1* with PKG inhibition samples were determined in Enrichr (https://maayanlab.cloud/Enrichr/), where differentially phosphorylated sites mentioned above were used as an input list to perform pathway-based analysis of Kyoto Encyclopedia of Genes and Genomes (KEGG).

### 4.5. Cryosection, Immunohistochemistry, and Proximity Ligation Assay (PLA)

Retinal tissues from *rd1* and wt in vivo at P9, as well as P11 cultured explants from *rd1*, were treated with 4% formaldehyde for 2 h, washed 3 × 15 min in phosphate-buffered saline (PBS), cryoprotected in PBS + 10% sucrose for overnight at 4 °C and subsequently with PBS + 25% sucrose for 2 h. After embedding in a medium with 30% bovine serum albumin (BSA; Cat. No.: A5253-250G; Sigma-Aldrich, St. Louis, MO, USA) and 3% gelatine (Cat. No.: 1040781000, Merck Millipore, Burlington, MA, USA) mixed in H_2_O, 12 μm thick retinal cross-sections were cut and collected from an HM560 cryotome (Microm, Walldorf, Germany). The sections were stored at −20 °C for later usage. Cryosections were then used for immunostaining and PLA. Totally 10 retinal samples were used, with 5 each in either *rd1* or wt strains, while 13 *rd1* explants were used, with 4 in the group of retinas with 4 days of PKG inhibition and 3 each in the other 3 groups.

For immunostaining, briefly, the cryosections were dried at room temperature for 15 min and rehydrated in PBS. Then they were blocked with 1% BSA + 0.25% Triton X100 + 5% goat serum in PBS at room temperature for 45 min. The primary antibody anti-*RAF1* (Cat#: MA5-17162, ThermoFisher) was diluted with 1% BSA and 0.25% Triton X100 in PBS (PTX) and incubated at 4 °C overnight; a no primary antibody control ran in parallel. Sections were washed 3 × 5 min each in PTX and incubated with a donkey anti-mouse IgG (H+L) highly cross-adsorbed secondary antibody, Alexa Fluor™ Plus 488 (#A32766, ThermoFisher) at 1:800 dilution in PTX. After 3 × 5-min PBS washes, the sections were mounted with Vectashield DAPI (Vector, Burlingame, CA, USA).

The PLA, which detects if two antigens are in close proximity (<40 nm) with each other, was performed on cryosections, using Detection Reagents Red kit (DUO92008-100RXN, Merck, Readington Township, NJ, USA), the PLA Probe anti-rabbit PLUS (DUO92002, Merck, Readington Township, NJ, USA), and anti-mouse MINUS (DUO92004, Merck, Readington Township, NJ, USA). The procedure followed the manufacturer’s instructions. In short, the retinal sections were blocked with blocking solution (provided in PLA Probe kit) for 45 min at 20 °C, and primary antibodies antibody anti-*RAF1* (Cat#: MA5-17162, ThermoFisher) and anti-Phosphoserine (ab9332, Abcam, Cambridge, UK) were incubated overnight at 4 °C. PLA Probe anti-rabbit PLUS, and anti-mouse MINUS were incubated for 1 hr at 37 °C. The ligation and amplification steps were performed using the Detection Reagents Red, followed by the addition mounting of Vectashield DAPI (Vector, Burlingame, CA, USA).

### 4.6. Microscopy and Image Processing

A Zeiss Imager Z1 Apotome Microscope (Zeiss, Oberkichen, Germany) with a Zeiss Axiocam digital camera was used for microscopy observations. Image generation and contrast enhancement were performed identically for all images via the ZEN2 software (blue edition). The immunostaining was analyzed for staining differences via three sections, each from three to five animals for each condition, after which the fluorescent intensities of positive cells were randomly distributed within the area of interest (outer nuclear layer, ONL, i.e., the photoreceptor layer) were assessed. Fluorescence intensity was captured and analyzed by the ImageJ software (version 1.53a, NIH, Rockville, MD, USA). The freehand selection function was used to target the ONL, after which the fluorescence intensity was calculated with the measure function. The values of all sections from the same animal were averaged. For PLA, the images were also generated with the aid of a Zeiss Imager Z1 Apotome Microscope and the ZEN2 software (blue edition). The punctuations of ONL, which represent reaction products from when the two antigens are in close proximity, were counted manually in three sections, each from three to five animals for each condition, with values from the same animal averaged. The experimenter was unaware of the groups of the samples during image processing.

### 4.7. Statistical Analysis

The Student’s *t*-test was used to compare the values of immunostaining intensities and PLA punctuations in different conditions in R, and the *p*-value cutoff was defined as 0.05. Visualization of KEGG analysis was also performed in R.

## Figures and Tables

**Figure 1 ijms-24-09836-f001:**
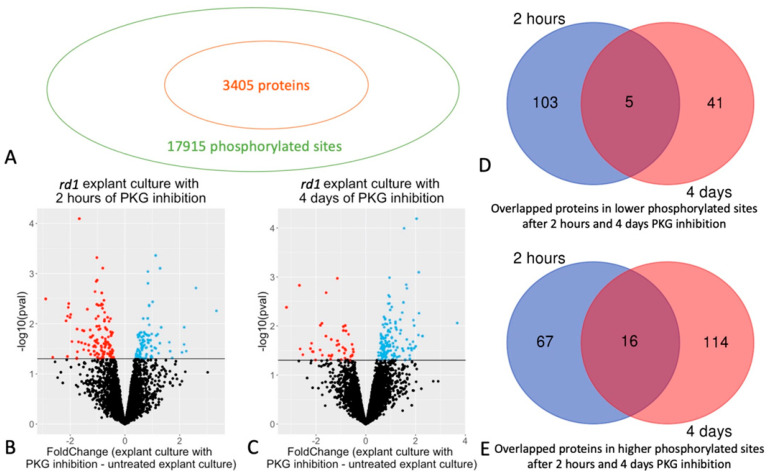
Phosphorylated sites and proteins in retinal explants affected by cGMP-dependent protein kinase (PKG) inhibition. Retinal explants generated from the *rd1* strain were treated with PKG inhibitor for 2 h or 4 days, and phosphoproteomics was done via mass spectrometry (MS) after phosphopeptide enrichment. (**A**) 17,915 phosphorylated sites of 3405 proteins were identified. (**B**) Volcano plot showing Fold Change between and -Log10(*p*-value) of phosphorylated sites with 2 h treatment. (**C**) Volcano plot showing Fold Change between and -Log10(*p*-value) of phosphorylated sites with 4 days treatment. For both B and C, red and blue dots represent sites with decreased or increased phosphorylation, respectively (*p* < 0.05), while black dots represent sites without difference. (**D**) Venn diagram showing the overlapped proteins in decreased phosphorylated sites after 2 h and 4 days PKG inhibition. (**E**) Venn diagram showing the overlapped proteins in increased phosphorylated sites after 2 h and 4 days PKG inhibition.

**Figure 2 ijms-24-09836-f002:**
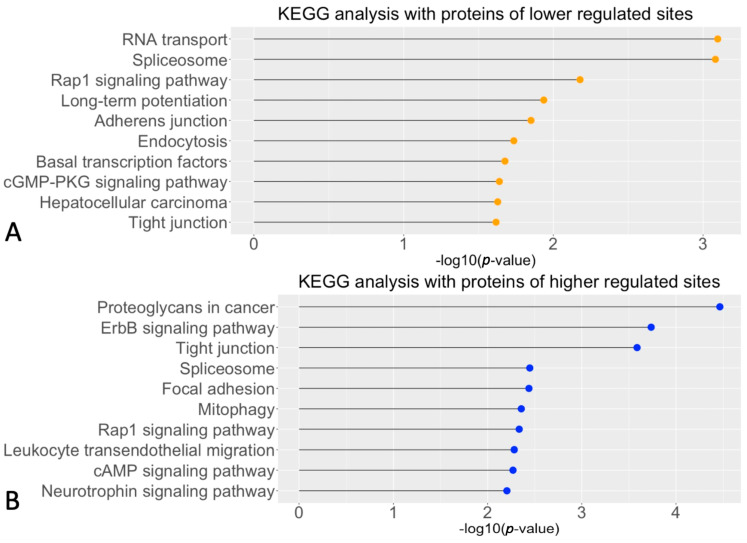
KEGG analyses of pathways connected with the observed phosphorylation changes. (**A**) Top 10 significant biological pathways enriched in the KEGG analysis and thus connected with proteins having decreased phosphorylated sites. (**B**) Top 10 significant biological pathways for sites with increased phosphorylation.

**Figure 3 ijms-24-09836-f003:**
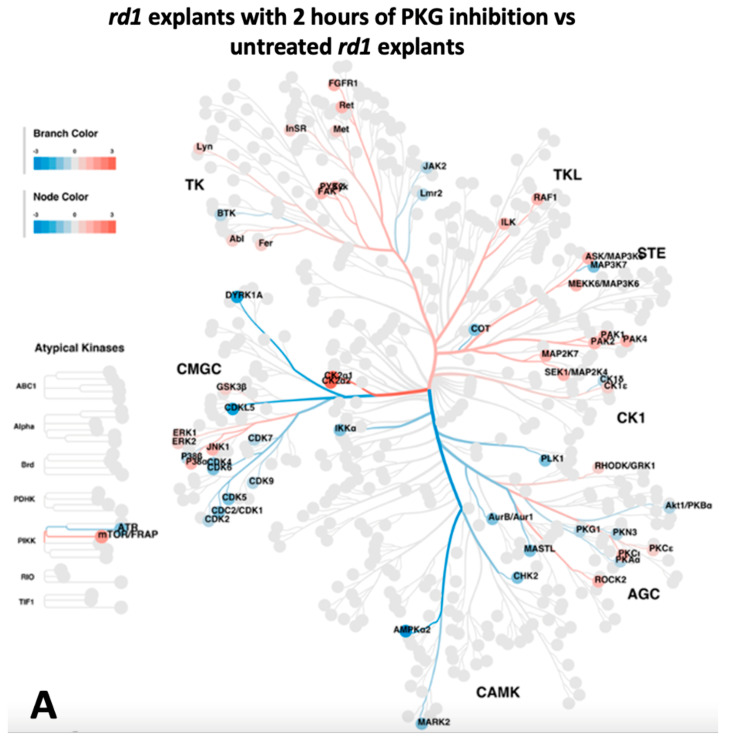
Potential altered kinases, as indicated by the upstream kinase tool, are visualized as kinome phylogenetic trees. The branch and node colors are encoded by Fold Change, with values <0 (in blue) and >0 (in red) representing kinase activity as decreased or increased, respectively, in *rd1* explants with PKG inhibition compared to untreated rd1 explants. (**A**) Kinome phylogenetic tree with potential kinases altered in *rd1* with PKG inhibition for 2 h. (**B**) Kinome phylogenetic tree with potential kinases altered in rd1 with PKG inhibition for 4 days.

**Figure 4 ijms-24-09836-f004:**
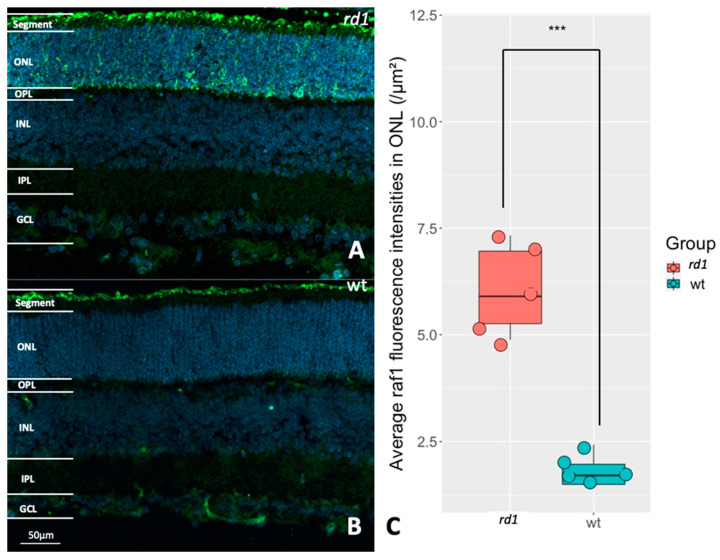
Evaluation of RAF proto-oncogene serine/threonine–protein kinase (*RAF1*) expression in *rd1* and wt retinas. (**A**): Immunostaining of *RAF1* (green) in a P11 *rd1* retina. (**B**): Immunostaining of *RAF1* (green) in a P11 wt retina. DAPI (blue) was used as a nuclear counterstain. (**C**): The box chart shows the comparison of RAF fluorescence intensities within the outer nuclear layer (ONL) between *rd1* (n = 5) and wt (n = 5), *** *p* < 0.001.

**Figure 5 ijms-24-09836-f005:**
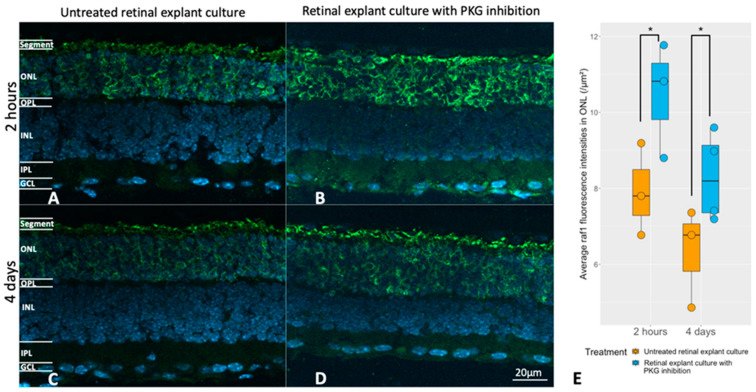
Evaluation of the relation between the cGMP-PKG system and *RAF1* expression. Retinal explants from *rd1* were treated with 50 μM Rp-8-Br-PET-cGMPS (PKG inhibitor) for 2 h (n = 3), and their untreated counterparts (n = 3) or the same compound in the same concentration for 4 days (n = 4) and their untreated controls (n = 3). (**A**,**B**) represent *RAF1* (green) immunostaining in untreated retinal explants and their counterparts with 2 h of PKG inhibition. (**C**,**D**) represent immunostaining of *RAF1* in untreated retinal explants and their counterparts with 4 days of PKG inhibition. DAPI (blue) was used as nuclear counterstain. (**E**) The box chart compares RAF fluorescence intensities within the ONL between untreated explants and peers with PKG inhibitors in different treatment lengths. * *p* < 0.05.

**Figure 6 ijms-24-09836-f006:**
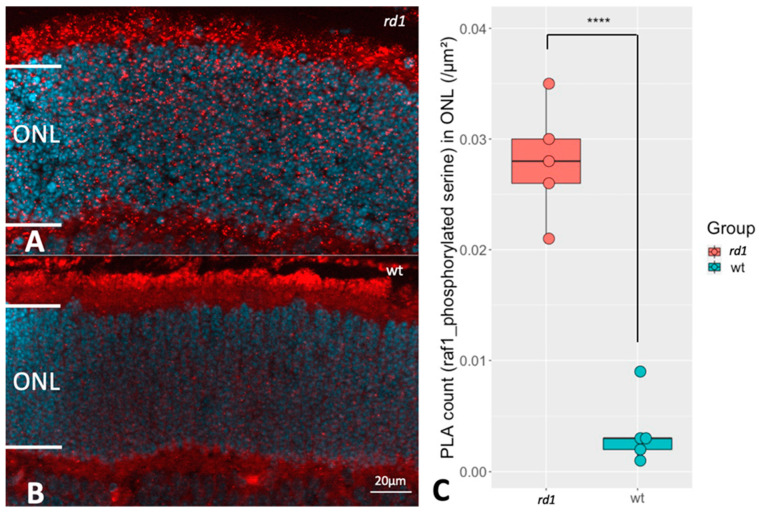
Comparison of *RAF1* phosphorylation between *rd1* and wt retinas. The PLA punctuations indicate that *RAF1* is in proximity with phosphoserine, which reveals the *RAF1* phosphorylation profile. (**A**,**B**) represent PLA punctuations (*RAF1* and phosphorylated serine) in ONL from *rd1* and wt, respectively. DAPI (blue) was used as nuclear counterstain (**C**) The box chart shows the comparison of PLA counts in ONL of retinas from rd1 (n = 5) and wt (n = 5). **** *p* < 0.0001.

**Figure 7 ijms-24-09836-f007:**
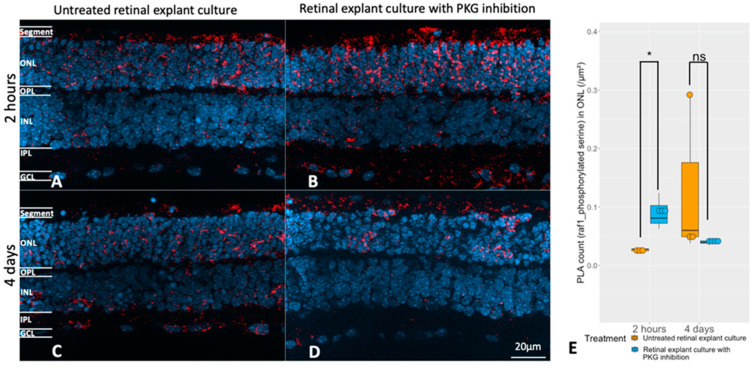
Evaluation of the relation between the cGMP-PKG system and *RAF1* phosphorylation. Retinal explants from *rd1* were treated with 50 μM Rp-8-Br-PET-cGMPS (PKG inhibitor) for 2 h (n = 3), and their untreated counterparts (n = 3) or the same compound in the same concentration for 4 days (n = 4) and their untreated controls (n = 3). (**A**,**B**) represent PLA (red, *RAF1*, and phosphorylated serine) in untreated retinal explants and the counterparts with 2 h PKG inhibition. (**C**,**D**) represent PLA punctuations in untreated retinal explants and the counterparts with 4 days PKG inhibition. DAPI (blue) was used as nuclear counterstain. (**E**) The box chart shows the comparison of PLA counts within ONL between untreated retinal explants controls and peers with PKG inhibitors in different lengths of treatment (n = 3–4). * *p* < 0.05.

**Table 1 ijms-24-09836-t001:** Lists of proteins showing decreased phosphorylation after PKG inhibition for either 2 h or 4 h, as well as reduced phosphorylation at both time points. The left and center columns only show the 20 proteins with the greatest fold change for 2 h and 4 days, respectively. The total number of proteins with decreased phosphorylation was 108 and 46 after 2 h and 4 days of PKG inhibition, respectively. Full lists of all proteins/peptides with decreased phosphorylation are found in Appendix A.

Top 20 after 2 h			Top 20 after 4 Days			Appearing in Both	
Gene Symbol	Protein Name	Fold Change	Gene Symbol	Protein Name	Fold Change	Gene Symbol	Protein Name
*CDC27*	cell division cycle 27	−2.894	*EPN1*	epsin 1	−3.188	*PKN1*	protein kinase N1
*AUTS2*	activator of transcription and developmental regulator	−2.638	*PKN1*	protein kinase N1	−2.668	*CIC*	capicua transcriptional repressor
*DMXL2*	Dmx like 2	−2.150	*PITPNM1*	phosphatidylinositol transfer protein membrane-associated 1	−2.640	*TOP2B*	DNA topoisomerase II beta
*PICALM*	phosphatidylinositol binding clathrin assembly protein	−2.100	*MAP7D1*	MAP7 domain containing 1	−2.530	*CCDC88A*	coiled-coil domain containing 88A
*TRAF7*	TNF receptor-associated factor 7	−2.100	*MTMR2*	myotubularin related protein 2	−2.225	*PBRM1*	polybromo 1
*PARD3B*	par-3 family cell polarity regulator beta	−2.069	*GRK1*	G protein-coupled receptor kinase 1	−2.180		
*AMPH*	amphiphysin	−2.051	*PCBP1*	poly(rC) binding protein 1	−2.144		
*ATG16L1*	autophagy related 16 like 1	−2.006	*JPT1*	Jupiter microtubule associated homolog 1	−2.115		
*RNF34*	ring finger protein 34	−1.980	*EPB41*	Protein 4.1 isoform 2	−2.023		
*ACIN1*	apoptotic chromatin condensation inducer 1	−1.934	*GPHN*	Gephyrin	−1.836		
*SYN3*	synapsin III	−1.759	*PPP6R2*	protein phosphatase 6 regulatory subunit 2	−1.826		
*DPY30*	dpy-30 histone methyltransferase complex regulatory subunit	−1.759	*FLNB*	filamin B	−1.762		
*ZNF687*	zinc finger protein 687	−1.747	*RNF20*	ring finger protein 20	−1.747		
*CTBP2*	C-terminal binding protein 2	−1.736	*TSC22D1*	TSC22 domain family member 1	−1.602		
*CCDC88A*	coiled-coil domain containing 88A	−1.696	*LIN37*	lin-37 DREAM MuvB core complex component	−1.595		
*SMPD3*	sphingomyelin phosphodiesterase 3	−1.666	*CIC*	capicua transcriptional repressor	−1.480		
*NKTR*	natural killer cell triggering receptor	−1.583	*SLC20A2*	solute carrier family 20 member 2	−1.416		
*NONO*	non-POU domain containing octamer binding	−1.554	*PRKG1*	protein kinase cGMP-dependent 1	−1.387		
*RBM8A*	RNA binding motif protein 8A	−1.509	*ARHGAP21*	Rho GTPase activating protein 21	−1.385		
*SNX2*	sorting nexin 2	−1.500	*PABIR1*	PP2A Aalpha (PPP2R1A) and B55A (PPP2R2A) interacting phosphatase regulator 1	−1.356		

**Table 2 ijms-24-09836-t002:** Lists of proteins showing increased phosphorylation after PKG inhibition for either 2 h or 4 h, as well as increased phosphorylation at both time points. The left and center columns only show the 20 proteins with highest fold change for 2 h and 4 days, respectively. The total number of proteins with increased phosphorylation was 83 and 130 after 2 h and 4 days of PKG inhibition, respectively. Full lists of all proteins/peptides with increased phosphorylation are found in Appendix A.

Top 20 after 2 h			Top 20 after 4 Days			Appearing in Both	
Gene Symbol	Protein Name	Fold Change	Gene Symbol	Protein Name	Fold Change	Gene Symbol	Protein Name
*HNRNPUL1*	heterogeneous nuclear ribonucleoprotein U like 1	3.343	*MAP1B*	microtubule-associated protein 1B	3.674	*HNRNPUL1*	heterogeneous nuclear ribonucleoprotein U like 1
*EEF1B2*	eukaryotic translation elongation factor 1 beta 2	2.593	*LGALS12*	galectin 12	2.279	*PALM2AKAP2*	PALM2 and AKAP2 fusion
*HNRNPU*	heterogeneous nuclear ribonucleoprotein U	2.242	*HNRNPUL1*	heterogeneous nuclear ribonucleoprotein U like 1	2.138	*DLG1*	discs large MAGUK scaffold protein 1
*GTPBP1*	GTP binding protein 1	2.165	*NR5A1*	nuclear receptor subfamily 5 group A member 1	2.128	*SEPTIN4*	septin 4
*MAP4*	microtubule associated protein 4	2.161	*GTF2F1*	general transcription factor IIF subunit 1	2.115	*MAP1B*	microtubule-associated protein 1B
*VPS53*	VPS53 subunit of GARP complex	2.083	*SRRM2*	serine/arginine repetitive matrix 2	2.041	*HNRNPU*	heterogeneous nuclear ribonucleoprotein U
*NES*	nestin	1.725	*RBSN*	rabenosyn, RAB effector	2.006	*MCM7*	minichromosome maintenance complex component 7
*LARP1*	La ribonucleoprotein 1, translational regulator	1.650	*TPD52L2*	TPD52 like 2	1.946	*ATF7IP*	activating transcription factor 7 interacting protein
*CDK13*	cyclin-dependent kinase 13	1.621	*VSX2*	visual system homeobox 2	1.945	*MAP4*	microtubule-associated protein 4
*AFDN*	afadin, adherens junction formation factor	1.323	*RBM12*	RNA binding motif protein 12	1.915	*ANK2*	ankyrin 2
*PXN*	paxillin	1.284	*DZIP3*	DAZ interacting zinc finger protein 3	1.897	*SRRM1*	serine and arginine repetitive matrix 1
*RCOR1*	REST corepressor 1	1.275	*ABLIM1*	actin binding LIM protein 1	1.842	*KIAA0930*	KIAA0930
*RUNX1T1*	RUNX1 partner transcriptional co-repressor 1	1.230	*PITPNM2*	phosphatidylinositol transfer protein membrane-associated 2	1.827	*CCDC88A*	coiled-coil domain containing 88A
*HTATSF1*	HIV-1 Tat specific factor 1	1.199	*ELF2*	E74 like ETS transcription factor 2	1.778	*CTBP2*	C-terminal binding protein 2
*ZC3H6*	zinc finger CCCH-type containing 6	1.143	*BICRAL*	BRD4 interacting chromatin remodeling complex associated protein like	1.771	*MAP2*	microtubule-associated protein 2
*BRAF*	B-Raf proto-oncogene, serine/threonine kinase	1.127	*PALM2AKAP2*	PALM2 and AKAP2 fusion	1.761	*RERE*	arginine-glutamic acid dipeptide repeats
*BASP1*	brain abundant membrane attached signal protein 1	1.122	*DNAJC5*	DnaJ heat shock protein family (Hsp40) member C5	1.707		
*PLCL2*	phospholipase C like 2	1.086	*PACS2*	phosphofurin acidic cluster sorting protein 2	1.705		
*SPHK2*	sphingosine kinase 2	1.012	*CCNL1*	cyclin L1	1.650		
*SEPTIN4*	septin 4	0.987	*EIF4EBP1*	eukaryotic translation initiation factor 4E binding protein 1	1.647		

**Table 3 ijms-24-09836-t003:** Kinases with proposed altered activities after various lengths of PKG inhibition, as indicated by the upstream kinase tool. The four columns give kinases that appear in both of the two situations indicated at the head of each column. Full lists of all proposed kinase alterations are found in Appendix A.

Proteins with Decreased Phosphorylation after PKG Inhibition for 2 h/Proteins with Decreased Phosphorylation after PKG Inhibition for 4 Days	Proteins with Increased Phosphorylation after PKG Inhibition for 2 h/Proteins with Increased Phosphorylation after PKG Inhibition for 4 Days	Proteins with Decreased Phosphorylation after PKG Inhibition for 2 h/Proteins with Increased Phosphorylation after PKG Inhibition for 4 Days	Proteins with Increased Phosphorylation after PKG Inhibition for 2 h/Proteins with Decreased Phosphorylation after PKG Inhibition for 4 Days
*CDK1*	*CSNK1E*	*CDK2*	*MAPK3*
*PRKACA*	*MAP2K7*	*PRKAA2*	*MAPK1*
	*MAP2K4*	*MAP3K7*	*PRKCE*
	*MAP3K5*	*CDK7*	
	*CSNK2A1*		
	*CSNK2A2*		
	*CSNK1E*		
	*BCR*		
	*MAPK8*		
	*MTOR*		
	*MAPK14*		

## Data Availability

The data presented in this study are available on request from the corresponding author.

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
