# Peer review of "The Phosphoproteome of the Rd1 Mouse Retina, a Model of Inherited Photoreceptor Degeneration, Changes after Protein Kinase G Inhibition"

_ijms, 2023, doi:10.3390/ijms24129836_

Round 1
Reviewer 1 Report
In the manuscript authors have investigated degenerating of the retina after protein kinase G inhibition using explant rd1 rodent model.
Open in the title rd1 and mention animals and explant based to the animals.
Introduction
-check text size. It changes at least in the introduction
Material and methods
- If 19 means the amount it should be (wt; n=19) instead of (wt; 19). As well the amount of male and females need to introduce.
- What means next sentence? ”and used irrespective of sex” Please, clarify gender.
- lines 87-89 are confusing;
” In parallel, another 4 groups of rd1 explants, reaching a total of 13 explants, were generated, with 4 such in the group of retinas with 4 days of PKG inhibition and 3 each in the other 3 groups”.
Please clarify and make easier to undestant the meaning of those groups and the treatments and the amount of samples/groups etc.
Results
Introduce little bit more results output of the study and highlight little bit more the key results. Do some collective sentence in the end of the particular result part as conclude the main findings.
- line 234, what means ”corresponding…” Does it mean control group? Need to clarify.
- Fig 1B and 1C need to be larger to see text as well better.
- Can table 1 and 2 be some how shorter and not so many pages? Those can be easier to read if example exclude not important results and highlight most important. Otherwise it is possible to include those as supplementary information and only introduce results shortly in the text?
- Add table 1 heading top of the table not end of it. As well other tables.
- Check if there is refered in the text to the table 3 before it appear. Always should firts refer and then show the table near text. Top of the table 3 is confusing? Decrease or Increase and 2 hour or 4 days?? Make clearer. or explain little bit in the table title at top of the table.
- Open abbreviations when mentioned first time example KEGG
- Figure 2, make it larger to see easier the text
- lines 316-317; mention in the text at least few of kinases which have opposite reactions in different time points.
- Figure 3, text need to be more visible.
- Explain reason for exclude other kinases and only RAF1 was investigated. Is it said quite much that this one kinase is critical for degeneration.
- lines 368-369 and 371-372, explain the mean of differences between 2 hour and 4 days. Does it affect to increase or decrease retinal degeneration that Raf phosphorylation and activity changed during timepoints. Is it beneficial or detrimental effect?
- lines 373-374, explain shortly what that means for degeneration of retina. Increase or decrease it?
- PLA as well need to open when mentioned first time. Please, go all abbreviations through.
Discussion
- Basicly discussion is wide but it was little bit confusing go it through with very detailed information related to some components which were not highlighted in the study results. RAF1 was more highlighted but it came in the end of the discussion. Please make discussion little bit clearer according to the results and modify little bit presenting of the results to match better to the discussion. Need to improve and may add little bit more background of the meaning. At least from beginning because after RAF1 introduction it was better.
- Discuss little bit related to retinitis pigmentosa. Now the disease background stay quite much to hide throughout the manuscript.
Manuscript need to go through and make some parts to more easy and understandable to readers.
Author Response
The authors thank the reviewer for the valuable comments, which all helped to improve the manuscript, and the responses are shown below:
Introduction
-check text size. It changes at least in the introduction
The text size in the Introduction, as well as the whole main text, has been made uniform in the revision.
Material and methods
If 19 means the amount it should be (wt; n=19) instead of (wt; 19). As well the amount of male and females need to introduce.
Thank you for addressing this confusing point. Here 19 means the reference number. We rephrased this sentence in the revision to avoid misunderstanding.
What means next sentence? ”and used irrespective of sex” Please, clarify gender.<
This sentence provides general information about how the animals were taken care of, and if there was a preference for using either males or females, which there was not. The animals for the experiment in this study were thus used unspecific to sex/gender (which is very hard to determine at the age of P5), just as we have done in the previous research in our group.
lines 87-89 are confusing;
” In parallel, another 4 groups of rd1 explants, reaching a total of 13 explants, were generated, with 4 such in the group of retinas with 4 days of PKG inhibition and 3 each in the other 3 groups”.
Please clarify and make easier to undestant the meaning of those groups and the treatments and the amount of samples/groups etc.
We agree that this phrasing was difficult to follow, and have now rephrased this sentence in the revision in lines 88-93 to make it clear, with the details of groups, treatment and sample size.
Results
Introduce little bit more results output of the study and highlight little bit more the key results. Do some collective sentence in the end of the particular result part as conclude the main findings.
In the revision, we added more concluding remarks in each part of the results to make it more readable.
line 234, what means ”corresponding…” Does it mean control group? Need to clarify.
Thanks for the careful reviewing which detected a phrasing mistake. The same information was duplicated in the last sentence, hence in the revision this sentence was deleted.
Fig 1B and 1C need to be larger to see text as well better.
A new figure 1 was provided in the revision with clearer font in the Figure 1B and 1C.
Can table 1 and 2 be some how shorter and not so many pages? Those can be easier to read if example exclude not important results and highlight most important. Otherwise it is possible to include those as supplementary information and only introduce results shortly in the text?
Thank you for this suggestion. We think, though, that many readers like to have direct and quick access to the identities of the altered proteins, and that this motivates the inclusion of such tables in the main paper. Still we in the revision shortened Table 1 and Table 2 to one page to make them more reader friendly. In addition we removed the columns for p-values, since this information is not critical here (but can be found in the supplementary material if needed) and thus we could make the tables even more easy to read.
Add table 1 heading top of the table not end of it. As well other tables.
All table headings were added on top of the tables in the revision.
Check if there is refered in the text to the table 3 before it appear. Always should firts refer and then show the table near text. Top of the table 3 is confusing? Decrease or Increase and 2 hour or 4 days?? Make clearer. or explain little bit in the table title at top of the table.
In the revision, the reference of Table 3 was changed, and the details can be seen in lines 354-355. The Table 3 appeared after it is referred to in the revised version. The content on top of Table 3 was changed to make it clearer.
Open abbreviations when mentioned first time example KEGG
The abbreviations are now appropriately used in the revision.
Figure 2, make it larger to see easier the text
A larger Figure 2 is now used, with an adjustment to make the font clearer.
lines 316-317; mention in the text at least few of kinases which have opposite reactions in different time points.
The revision provides this content, and the details can be seen in lines 359-360.
Figure 3, text need to be more visible.
The Figure 3 has in the revision been split into Figure 3A and Figure 3B, and the text should now be more visible.
Explain reason for exclude other kinases and only RAF1 was investigated. Is it said quite much that this one kinase is critical for degeneration.
In the original text, we mentioned the reason that RAF1 has a specific expression in photoreceptors, which is likely to have a direct relation with retinal degeneration. While other kinases may also function during photoreceptor death, they were however excluded here because they are not specifically expressed within photoreceptors. Hence other targets may certainly exist, and as such deserve further investigation in the coming days but were not addressed in this study.
lines 368-369 and 371-372, explain the mean of differences between 2 hour and 4 days. Does it affect to increase or decrease retinal degeneration that Raf phosphorylation and activity changed during timepoints. Is it beneficial or detrimental effect?
lines 373-374, explain shortly what that means for degeneration of retina. Increase or decrease it?
In the revision, more statements were made to address these comments, with the details seen in lines 436-440. In general, the data indicates that RAF1 activities were affected by different lengths of PKG inhibition. According to the results of this study, it is definitely so that RAF1 functions in ways related to photoreceptor degeneration, but at this point the mechanism cannot be precisely established. It would therefore be unwise to claim either a neuroprotective or a neurodegenerative effect of RAF1 in retinal degeneration, since more experiments are needed to come to a clear conclusion.
PLA as well need to open when mentioned first time. Please, go all abbreviations through.
All abbreviations have been appropriately used in the revision.
Discussion
Basicly discussion is wide but it was little bit confusing go it through with very detailed information related to some components which were not highlighted in the study results. RAF1 was more highlighted but it came in the end of the discussion. Please make discussion little bit clearer according to the results and modify little bit presenting of the results to match better to the discussion. Need to improve and may add little bit more background of the meaning. At least from beginning because after RAF1 introduction it was better.
We agreed that the basic discussion is wide, hence we made the changes profoundly in the Discussion part in the revision. To clearly present the results, we shortened the Discussion with respect to substrates and kinases regulated by cGMP-PKG. More background content was added to make it more readable.
Discuss little bit related to retinitis pigmentosa. Now the disease background stay quite much to hide throughout the manuscript.
The background of Retinitis pigmentosa was added in the Discussion part of the revision, with the details seen in lines 489-498.
Reviewer 2 Report
The study is very interesting, relevant, and of general interest to the readers to the readers of this Journal’s Special Issue of “Retinal Diseases and Cell Signaling”; Section of Molecular Neurobiology.
This study aims to manipulate the cGMP- PKG (protein kinase G) system through PKG inhibition by a cGMP-analogue, in retinal pathologies.
The main novelty factor is identifying a host of new potential cGMP-PKG downstream substrates and related kinases, providing more insights into the retinal degeneration mechanisms in Retinitis pigmentosa.
We found this article well written, with a good organization of the contents.
The introduction briefly covers old and new references and perfectly integrates the theme's main aspects.
The core experimental design of the “Organotypic retinal explant culture”, “Sample preparation for MS”, “MS acquisition and analysis”, “Cryosection, immunohistochemistry and Proximity ligation assay (PLA)” and correspondent statistical analysis to support the conclusions, were carefully elaborated and meticulously presented.
Regarding the discussion of the results, we found it suitable. A very nice graphic pics/graphs accompanying the discussion increase the understanding of the discussed theme and clarify the reasoning. We congratulate the authors on that!
Specific comments:
#1_ L37-38+ L261-266+L267-272+L273-277_ The text appears to have been unformatted.
#2_ The references need a careful inspection for uniformity.
Author Response
The authors thank the reviewer for the valuable comments, and the responses are shown below:
#1_ L37-38+ L261-266+L267-272+L273-277_ The text appears to have been unformatted.
The text mentioned have been formatted in the revision, and the details can be seen in L37-38, L281-283, L301-303 and L374-375.
#2_ The references need a careful inspection for uniformity.
We went through the references and made necessary changes to confirm the uniformity in the revision.
Round 2
Reviewer 1 Report
Authors have improved manuscript quite well according to recommendations. There is still few points gave to mind and should as well revise. It would as well be nice to see Figures as final version without old figures.
Supplementary material
- Label supplementary tables the way that reader can read/understand what those present without text after opening.
Title
Title could be improved and ”..degenerating rd1 retina..” should explain little bit more with few words e.. animal species and open rd1
Material and methods
- Number of animals should still introduce and the amount of male and female animals. If it is hard to explain the gender, it could as well open little bit for reader at least with one short sentence why not.
Results
- Please, move all text under the table into top of the table after title.There should not be text under table or only definition of abbreviations, not anything else. Table text should be top of the table.
- -Figure two, please take small figures away and left only bigger ones to see the final version better.
- Please, check carefully the group labeling of the table 3 if they are same or different (e.g. second and fourth groups).
- Figure 3 take small pictures away.
Discussion
- I still would like to see the discussion as cleaned final version without modifications appear. To see it better.
Need to proofread after final cleaned modified version is ready without track changes visible.
Author Response
The authors thank the reviewer for the informative updated comments. According to the MDPI policy, the “track function” was applied in the revised manuscript to mark up any changes. We are of course happy to provide a clean version, which may address some of the issues. The responses are shown below:
Supplementary material
- Label supplementary tables the way that reader can read/understand what those present without text after opening.
Here we are not quite certain what the reviewer wants, but we have now re-labelled the supplementary files, such that the file names indicate what sort of data they present.
In addition, we have increased the text size of the descriptions at first row of each file and sheet. We hope that this will make it easier to see what he various files contain. In addition, we have increased the text size of the descriptions at first row of each file and sheet. We hope that this will make it easier to see what he various files contain. We will, of course, be happy to make further adjustments to these tables if our interpretation of the reviewer’s sugggestion was not correct.
Title
Title could be improved and ”..degenerating rd1 retina..” should explain little bit more with few words e.. animal species and open rd1
In this revision, the title was changed to “The phosphoproteome of the rd1 mouse retina, a model of inherited photoreceptor degeneration, changes after protein kinase G inhibition”.
Material and methods
- Number of animals should still introduce and the amount of male and female animals. If it is hard to explain the gender, it could as well open little bit for reader at least with one short sentence why not.
Such information was added in this revision, and the details can be seen in Lines 76-78.
Results
- Please, move all text under the table into top of the table after title.There should not be text under table or only definition of abbreviations, not anything else. Table text should be top of the table.
The text parts under the tables were moved into top of the tables in the revised version.
- -Figure two, please take small figures away and left only bigger ones to see the final version better.
In this clean version, only the bigger figure was available.
- Please, check carefully the group labeling of the table 3 if they are same or different (e.g. second and fourth groups).
Thanks to the reviewer for the careful check, the mistake was corrected in the revision.
- Figure 3 take small pictures away.
Only bigger figures are present in the revision.
Discussion
- I still would like to see the discussion as cleaned final version without modifications appear. To see it better.
In this clean version, it is easier for the reviewer to read. The changes were seen in lines 437-446, lines 468-471, lines 480-481 and lines 489-491.